# Comparative Metabolomics of Early Development of the Parasitic Plants *Phelipanche aegyptiaca* and *Triphysaria versicolor*

**DOI:** 10.3390/metabo9060114

**Published:** 2019-06-13

**Authors:** Kristen Clermont, Yaxin Wang, Siming Liu, Zhenzhen Yang, Claude W. dePamphilis, John I. Yoder, Eva Collakova, James H. Westwood

**Affiliations:** 1School of Plant and Environmental Sciences, Virginia Polytechnic Institute and State University, Blacksburg, VA 24061, USA; kristc1@vt.edu; 2Department of Plant Sciences, University of California, Davis, CA 95616, USA; yaxwang@ucdavis.edu (Y.W.); sigliu@ucdavis.edu (S.L.); jiyoder@ucdavis.edu (J.I.Y.); 3Department of Biology, Pennsylvania State University, University Park, PA 16802, USA; zhenzhenyang.bing@hotmail.com (Z.Y.); cwd3@psu.edu (C.W.d.)

**Keywords:** parasitic plant, heterotrophy, *Phelipanche aegyptiaca*, *Triphysaria versicolor*, central carbon and nitrogen metabolism

## Abstract

Parasitic weeds of the family Orobanchaceae attach to the roots of host plants via haustoria capable of drawing nutrients from host vascular tissue. The connection of the haustorium to the host marks a shift in parasite metabolism from autotrophy to at least partial heterotrophy, depending on the level of parasite dependence. Species within the family Orobanchaceae span the spectrum of host nutrient dependency, yet the diversity of parasitic plant metabolism remains poorly understood, particularly during the key metabolic shift surrounding haustorial attachment. Comparative profiling of major metabolites in the obligate holoparasite *Phelipanche aegyptiaca* and the facultative hemiparasite *Triphysaria versicolor* before and after attachment to the hosts revealed several metabolic shifts implicating remodeling of energy and amino acid metabolism. After attachment, both parasites showed metabolite profiles that were different from their respective hosts. In *P. aegyptiaca*, prominent changes in metabolite profiles were also associated with transitioning between different tissue types before and after attachment, with aspartate levels increasing significantly after the attachment. Based on the results from ^15^N labeling experiments, asparagine and/or aspartate-rich proteins were enriched in host-derived nitrogen in *T. versicolor*. These results point to the importance of aspartate and/or asparagine in the early stages of attachment in these plant parasites and provide a rationale for targeting aspartate-family amino acid biosynthesis for disrupting the growth of parasitic weeds.

## 1. Introduction

A parasitic plant is a plant that derives some or all of its nutrients from another living plant through specialized feeding structures called haustoria. Parasitic plants are diverse, and vary in the extent to which they depend on the nutrients of their host. Hemiparasites retain the ability to photosynthesize, whereas holoparasites lack this ability and depend on their hosts for all nutrients. Facultative parasites are able to complete their lifecycle without a host, if necessary, while obligate parasites are absolutely reliant on a host. All Orobanchaceae species are parasitic on host plant roots, but species differing in host dependence may use different mechanisms to obtain resources. For example, holoparasites must acquire both carbon and nitrogen from their hosts, so these species make connections to both host xylem and phloem and are primarily phloem-feeding. In contrast, hemiparasites are primarily xylem-feeding and mainly take nitrogen from the host plant [1]. Studying parasitic plant metabolism has been one approach for identifying processes that may be targeted for disruption to improve crop resistance to parasitism [2,3,4].

Several species in the family Orobanchaceae are important weeds, and in particular, members of the genera *Phelipanche*, *Orobanche*, and *Striga* cause major agricultural damage. Weedy Orobanchaceae tend to thrive in warm, arid climates and grow primarily in Africa, the Mediterranean, and the Middle East [5]. It is estimated that 4%–5% of the world’s arable land is threatened in capacity for crop production by parasitic Orobanchaceae [6] and these parasites can cause up to 100% loss of crop yield [7]. These parasites are also prolific producers of small seeds; millions of new seeds per hectare can be added to the seed bank each year in fields where susceptible crops are grown. These parasitic plants are more challenging to control than non-parasitic weeds because they cause crop damage while they are still underground, where they are not accessible to mechanical control measures. Chemical control is also challenging because herbicides cannot selectively kill the parasite without harming the crop, and the non-photosynthetic parasite species are not susceptible to herbicides that disrupt chlorophyll production or photosynthesis [8]. 

The fact that parasitic plants are metabolically dependent on their hosts raises the possibility that reliance on host metabolism is a weakness that may be exploited for parasitic weed control. For parasite species that acquire carbon and nitrogen in organic forms, parasitism would seem to relieve them of the metabolic burden of incorporating carbon and nitrogen from inorganic materials. In Orobanchaceae species, a number of ammonium-assimilation-related enzymes have low activity, such as nitrate reductase (NR), glutamine synthetase (GS), and aspartate amino transferase (AAT) [9]. In addition, some members of these species are missing GS2, one of the two copies of GS that are normally found in higher plants [10,11]. This may be due to reduced selection pressure, given the role of GS2 in re-assimilating ammonia produced by photorespiration [12,13,14]. Free-living plants typically have equal or higher GS2 than GS1 activities. Within Orobanchaceae, the ratio of activities of the two isozymes depends loosely on the level of parasitism of the parasites studied: Obligate parasites primarily have GS1 activity—with holoparasites generally lacking GS2 activity and hemiparasites having low levels of GS2 activity relative to GS1 [10,11]. 

A corollary to the hypothesis that parasitic plants are deficient in aspects of their own metabolism is that parasitic plants would then adopt major aspects of the host metabolism because they would acquire host metabolites. However, recent reports contribute to a consensus that parasitic plants are largely self-regulating in metabolism [4,15,16], showing metabolite profiles that are distinct from their hosts, likely to support their parasitic lifestyles. A number of studies have compared metabolite profiles of the host root to Orobanchaceae holoparasites [3,15,17,18]. Limitations of these studies were in focusing on a single developmental stage and using obligate parasites with well-established attachments, thus ignoring the role of parasite development and generally neglecting facultative parasites. Many questions remain unanswered regarding the independence of parasite and host metabolisms in the context of parasite development during the transition from free-living to parasitic life stages. More attention should also be given to understanding differences between parasite species and their specific host interactions. 

The objective of this research was to fill these knowledge gaps by characterizing parasitic plant metabolism across key developmental stages in their life cycle and by comparing related species that contrast in level of host dependence. For the comparison of developmental stages, we focused on the transition from free-living to host-dependent stages of their life cycles. The free-living stage occurs in early development when the parasites must grow and develop either autotrophically (for facultative parasites) or as seedlings using only seed-storage reserves (for holoparasites). The process of host attachment is therefore pivotal in that this transition marks the switch from independent metabolism to host-fed metabolism and comprises the essence of parasitism. For evolutionary comparisons, we used the facultative hemiparasite *Triphysaria versicolor* and the obligate holoparasite *Phelipanche aegyptiaca*, both members of the family Orobanchaceae, the only parasitic plant family that includes species that span all levels of host dependence [19]. Our goal was to determine whether these species would show evidence of shared metabolic processes that could indicate a cross-cutting strategy for parasite metabolism. Although we used different hosts for each parasite species (justified in more detail in [20]), we were able to contrast the different parasite species with their hosts and with each other to gain insight into the extent to which parasite metabolism is self-regulating. 

Our data reveal the unique metabolic makeup of two representative Orobanchaceae species that differ in parasitic dependency and clarify the differences in feeding strategies during developmental transitions in these parasites. We show that nitrogen assimilation dynamics differ among the parasites and suggest that multiple metabolic mechanisms exist to enable successful parasitism.

## 2. Results

### 2.1. Parasitic Plants and Hosts Have Distinct Metabolite Profiles

Metabolite profiles were analyzed for two parasitic plant species and their associated hosts (Figure 1). The parasites differ in levels of host dependence and across life stages that span from host-independent to host-dependent stages. Steady-state levels of 28 metabolites were quantified in 16 sample types from parasites and their hosts. Four to five biological replicates of pooled parasite samples were analyzed for each sample type. When all stages of the two parasite and two host species were compared on a combined basis, each species demonstrated a distinctive metabolite profile as determined by principal component analysis (PCA) (Figure 2) and analysis of variance (ANOVA) (Appendix A). Thus, the two parasite species *P. aegyptiaca* and *T. versicolor* differed from each other in metabolite profiles, as did the parasite species and their respective hosts. The first three principal components (PCs) together accounted for 65% of the variance among samples. 

To gain a higher resolution picture of the factors contributing to differences between hosts and parasites and among different stages of parasite development, each parasite species was compared across stages and directly against its host. For *P. aegyptiaca*, the PCA and ANOVA indicate that metabolite profiles are distinguishable by developmental stage (Appendix A), with pronounced differences separating stages before and after vascular linkage to the host (stages 1–3 vs. stages 4.1–4.2) (Figure 3A). The parasite metabolite profile is also distinguishable from that of its host at all stages, and in fact appears more similar to the host prior to rather than after attachment to the host (Figure 3A). The PC 1 separates pre- from post-vascular *P. aegyptiaca*, PC 2 separates stages 4.1 and 4.2, and PC 3 separates pre-vascular *P. aegyptiaca* from *Arabidopsis thaliana* roots. Metabolites that account for the separation between samples are shown in the loading plots (Figure 3B). Prominent examples of differences in key metabolites include higher levels of γ-aminobutyric acid (GABA) and glucose in *A. thaliana* roots than in parasites, whereas parasites have more mannitol, proline, and lysine. *P. aegyptiaca* has higher levels of proline and lysine shortly after vascular attachment than it has after development of adventitious roots on the tubercle. The levels of many specific metabolites increased in *P. aegyptiaca* after its attachment to the host (stages 4.1 and 4.2) (Figure 4, Figure 5, and Appendix A). These metabolites include asparagine, glutamine, aspartate, glutamate, valine, and all detected sugars, sugar alcohols, and carboxylic acids except for glucose and mannitol. Generally, the attached parasites have relatively high levels of these compounds in comparison to both pre-attached *P. aegyptiaca* seedlings and to the host roots. The sum of the detectable proteogenic free amino acids (FAAs) is highest at stage 4.1 immediately after attachment, with a subsequent decrease at stage 4.2, although their levels remain higher than at the pre-attachment stages (Figure 4). Asparagine and glutamine levels peak at stage 4.1, while aspartate and glutamate levels are high at both stages 4.1 and 4.2. On the host side, *A. thaliana* root metabolite profiles are unaffected by infection with *P. aegyptiaca* (Figure 3A).

The PCA of *T. versicolor* on *Medicago truncatula* illustrates a different pattern of metabolism than seen in *P. aegyptiaca*, though some similarities exist. *T. versicolor* growth stages are not clearly separated by PCA of the metabolite profiles within the species, unlike growth stages in *P. aegyptiaca* (Figure 3C). However, the levels of compounds such as mannitol, fumarate/maleate, glutamate, and, to a lesser extent, malate and aspartate are higher in *T. versicolor* than in *M. truncatula* roots, which is consistent with differences observed in the *P. aegyptiaca*–*A. thaliana* system. ANOVA results corroborate the PCA results, as all *p*-values for all tested metabolites with growth stage as a factor were found statistically insignificant, as they exceeded the value of 0.05. (Appendix A). Only a few specific amino acids showed altered levels pre vs. post attachment, and proline was notable for an increase in stage 4 relative to earlier stages (Figure 4). On the host side, the *M. truncatula* roots were enriched in asparagine, valine, threonine, phenylalanine, and histidine relative to the parasite, with asparagine being the most abundant amino acid in the *M. truncatula* roots (Figure 3C,D, Figure 4 and Appendix A). Asparagine levels were constant through the first three stages, but decreased to 25% of the average of its previous levels in stage 4, after parasite attachment. Histidine, threonine, isoleucine, and phenylalanine also followed this pattern, and the trend was reflected in the sum of the detectable proteogenic FAAs (Figure 4 and Appendix A). GABA was the only metabolite that showed increased levels in the host root during *T. versicolor* parasitism (stages 3 and 4; Appendix A). Levels of the sugars, carboxylic acids, and sugar alcohols remained stable in the host root in response to parasitism. 

### 2.2. *T. versicolor* Preferentially Accumulates Asparagine/Aspartate

The relatively constant levels of amino acids across stages spanning pre- and post-attachment in *T. versicolor* provided limited insight into nitrogen assimilation. A further experiment was conducted to determine the compounds to which nitrogen from the host is directed first in *T. versicolor*. A split root system was used to feed ^15^N-labeled nitrate to the host without allowing direct uptake by the parasite (Figure 6) and protein-derived amino acids were analyzed in terms of levels/composition and ^15^N enrichment. In this type of experiment, when a single ^15^N substrate is provided (in this case at low isotopic enrichment), all N-containing compounds will be labeled to the same (low) degree within a given system after time periods exceeding rates of metabolism. As such, protein-derived amino acids in *M. truncatula* roots showed low, about 3% isotopic enrichment in analyzed amino acids regardless of parasitism when ^15^N nitrate at 5% enrichment was fed to half of the roots for several days (Mt-Uninfected and Mt-Infected, Figure 7A). Host root protein levels and amino acid composition of the total proteins were also unaffected by parasite attachment (Figure 7B,C).

*T. versicolor* feeding on *M. truncatula* roots showed low enrichment in most amino acids (about 1.5% ^15^N), confirming the uptake of nitrogen-containing compounds from the host. Interestingly, asparagine and/or aspartate (Asx; these two amino acids are indistinguishable due to acid-enabled hydrolysis of the amide group from asparagine during protein hydrolysis) showed a small, but statistically significant increase in labeling over other amino acids (Tv-Attached, Figure 7A). This increase in Asx ^15^N enrichment was specific to the attached parasite and was not observed in the host roots. Asx proportion in proteins was also increased (doubled) upon attachment to the host root (Tv-Attached, Figure 7B), while the composition of the remaining analyzed amino acids was not affected in *T. versicolor* attached to the host root (Tv-Attached vs. Tv-Water, Figure 7B). 

### 2.3. Nitrogen Assimilation is a Priority in Early Development of Parasitic Plants 

Two key genes for ammonium assimilation in plants are asparagine synthases (AS) and aspartate amino transferases (AAT), both of which occur in multigene families [22,23,24]. We searched the transcriptome data of the PPGP [21] to identify sequences coding for the *P. aegyptiaca* and *T. versicolor* versions of the corresponding transcripts. Two forms of AS and four forms of AAT were found, and the expression of all forms was inferred from RNA-seq data [21]. RNA-seq contigs used are listed in Appendix A. For *P. aegyptiaca*, expression of at least one form of each gene peaks early in development, around stage 2 when the haustorium is being initiated (Figure 8). Other forms of these genes have maximum expression around stages 3 and 4.1. For *T. versicolor*, AS expression is dominated by a single gene with high expression at stage 3 as well as stages 6.1 (vegetative stems/leaves) and 6.2 (flowers). Expression of AAT in *T. versicolor* is highest in seeds, but otherwise characterized by multiple forms expressed steadily throughout development. 

## 3. Discussion

This study aimed at understanding parasitic plant development and evolution as informed by metabolite profiling. We compared the metabolite profiles of the facultative parasite *T. versicolor* and its host *M. truncatula* with the obligate holoparasite *P. aegyptiaca* and its host *A. thaliana*. The focus was on the developmental stages that span the transition from free-living to parasitic in order to determine the impact of host-derived nutrition on the parasite. We have previously characterized these species with respect to transcriptomes [21,25,26], but metabolite profiles provide a fresh perspective on these questions. 

### 3.1. Metabolic Autonomy in Parasitic Plants

Our data show the metabolic profile varies greatly between the parasite and host species studied. Total detectable proteogenic FAA levels in *T. versicolor* were approximately half those of the host *M. truncatula* until the levels in the host decrease following vascular attachment, whereas total detectable proteogenic FAA levels in *P. aegyptiaca* were approximately twice those of *A. thaliana* roots after attachment to the host (Figure 4). The higher FAA levels in a holoparasite species than in its host is in agreement with what was reported for below ground *P. aegyptiaca* shoots parasitizing *Solanum lycopersicum* [15] and in older tubercles (roughly stage 4.2) of *Phelipanche ramosa* grown on *Brasica napus* [17]. Because neither parasite species adopted the metabolite profile of its host root after attachment, our data support the idea that parasites maintain autonomy over their primary metabolism (Figure 2 and Figure 3).

Another question is whether the two parasite species are similar in their metabolic compositions. *P. aegyptiaca* and *T. versicolor* are related parasite species that share at least a core set of gene functions involved in haustorial development and establishment on hosts [21]. Their common need to draw resources from the host could be hypothesized to lead to similar metabolite profiles in the two species, but this was not the case as the metabolite profiles of each species generally differed at all stages of growth (Figure 2). The differences between the two parasite species were most evident during the transition from free-living to host-feeding stages. *P. aegyptiaca* showed a marked shift in metabolite profiles pre- and post-attachment to the host, whereas *T. versicolor* did not. As an obligate holoparasite, *P. aegyptiaca* is not able to grow without a host and completely relies on influx of host-derived metabolites such as sugars and amino acids. Therefore, the peak in total detectable proteogenic FAAs at stage 4.1 (Figure 4) likely reflected a strong influx of an unknown form of nitrogen from the host following access to host-derived nutrients. Similarly, these tubercles (stages 4.1 and 4.2) also had substantial general increases in sugar, sugar alcohol, and carboxylic acid levels, except for glucose and mannitol (Figure 5 and Appendix A). However, the specific host-derived metabolites that are transported to these tubercules remain to be uncovered. In contrast, the *T. versicolor* metabolite profiles were not substantially changed following the connection of a haustorium to the host (Figure 3C and Figure 4). These differences between the developmental transitions of the two parasite species are likely due to the nature of parasite development post-attachment (Figure 3 and Figure 4). The development of lateral haustoria in *T. versicolor* is followed primarily by additional root growth, and the roots surrounding the haustorium do not transition into a different type of plant organ after haustorial attachment [27]. In contrast, the *P. aegyptiaca* developmental transition from a seedling to a tubercle represents a substantial change in tissue types as well as nutrient source [19]. 

We conclude that *P. aegyptiaca* and *T. versicolor* undergo shifts in their metabolite profiles that are more closely related to patterns of parasite development than to influences from host metabolism. This fits nicely with results of a study by Nativ et al. [28] of *P. aegyptiaca* metabolite profiles that spanned stages of development ranging from the tubercle (older than our stage 4.2) to mature shoots, and in organs ranging from roots to flowers. This work showed that the greatest differences in metabolite profiles were between different organs, rather than across time in the same organs. Although this study did not include seedling or very young tubercle stages (as we present in the current work), taken together our data show that *P. aegyptiaca* metabolite profiles differ primarily between major life stages and organs. From this perspective, the lack of metabolite profile change in *T. versicolor* is expected because pre- and post-parasitizing roots are both essentially root organs. 

### 3.2. Carbon Metabolism

The holoparasite *P. aegyptiaca* relies exclusively on the host for carbon resources, with a wide range of sugars, sugar alcohols, and carboxylic acids increasing after attachment to its host. In particular, increases in sugars such as fructose, glucose-6-P, and sucrose (Figure 5) reflect this phloem-feeding nature of the parasite. The increase in levels of the TCA cycle intermediates fumarate, malate, and citrate (Figure 5 and Appendix A) may indicate an acceleration in energy metabolism corresponding to a rapid increase in growth. The levels of these compounds do not change in *T. versicolor* after attachment to its host, perhaps reflecting the lower host dependence and the steadier continuation of growth of *T. versicolor* before and after haustorial attachment. 

One aspect of parasite metabolism that is conserved between the two species is mannitol accumulation. Parasites of the family Orobanchaceae convert hexoses into mannitol in order to increase sink strength [29,30]. Mannitol is a major metabolite for these parasites and may be a conserved metabolic strategy within Orobanchaceae as it has been detected in genera across the family [31]. Mannitol serves as a compatible solute, meaning it can accumulate within the cell without disrupting cell function [32]. Our results confirm that mannitol is specific to the Orobanchaceae species in these systems (Figure 5). Although trace amounts were detected in some of the host samples, it was not found in host roots that had no parasites, so this likely represents either backflow of the metabolite into the host or contamination due to small amounts of tissue from the haustorium embedded in the host root. In *P. aegyptiaca*, a reproducible, statistically significant decrease in mannitol levels at stage 4.1 was observed in comparison with other stages (Figure 5). This is opposite to the patterns seen in *P. ramosa* on *S. lycopersicum* [29], in which mannitol was higher in young tubercles (comparable with stage 4.1 in the current study) vs. germinated seeds and mature tubercles (comparable with stages 1 and 4.2, respectively).

### 3.3. Nitrogen Assimilation

The acquisition of nitrogen is a dominant theme in studies of parasitic plant metabolism [1,31] and our study confirms that parasitic plants rapidly accumulate amino acids involved in ammonium assimilation. Glutamine is the amino acid present at the highest levels in *A. thaliana* phloem and xylem sap [33,34], so the spike in glutamine levels in the parasite immediately after vascular attachment (Figure 4) may reflect the high levels of glutamine available for influx. Other studies indicate that *Phelipanche* species preferentially accumulate glutamine [17], along with glutamate, asparagine, and aspartate [15,18]. Although *T. versicolor* metabolite profiles were not affected by the attachment of the parasite to the host, the observed ^15^N labeling of amino acids and increased total protein levels after attachment imply that this facultative parasite is able to direct host-derived nitrogen into proteins without changing the steady state pools of FAA (Figure 7). 

The increase in the levels and ^15^N enrichment in protein-derived Asx in comparison with other amino acids raises the question of why Asx is not labeled to the same degree as other amino acids in feeding *T. versicolor* when the host-root Asx is labeled to the same extent as other amino acids. The levels and labeling of other observed amino acids did not change upon parasite attachment. A plausible explanation is that, due to preferential Asx uptake or metabolism, higher levels of organic ^15^N from the host get mixed with the original non-labeled Asx from the parasite than for other amino acids, which leads to the observed increase in labeling over other amino acids in the attached parasite. If the original non-labeled Asx levels were negligible in the parasite, the labeling in Asx would theoretically become similar to that in the host roots. Based on the increase in the levels and ^15^N enrichment in protein-derived Asx in comparison with other amino acids, it is likely the Asx is incorporated into an unknown Asx-rich protein. Collectively, both increased levels and labeling in Asx corroborate increased or preferential uptake and/or metabolism of Asx during facultative parasitism. These results point to the importance of Asx in post-attachment nitrogen metabolism of a facultative parasite.

Asparagine synthetase is considered an important enzyme for nitrogen assimilation and balancing the carbon to nitrogen ratios in Orobanchaceae. For example, expression of the AS gene in *T. versicolor* is also known to be upregulated by exposure to host root exudates [35] and it has been characterized in the related parasite, *Striga hermonthica* [16,36]. Our data also point to an increase in aspartate levels in *P. aegyptiaca* after attachment to the host (Figure 4). Aspartate represents 15% of the total detectable proteogenic FAAs in the parasite at stage 4.1 and 30% at stage 4.2. The role of Asx as metabolic intermediates in processing nitrogen is probably important considering that *A. thaliana* phloem contains less aspartate than glutamate [34]. While AS is important in holoparasite shoot development and in *T. versicolor* roots, we hypothesize that AAT plays a key role in amino acid metabolism in the holoparasite *P. aegyptiaca* during early development. AAT functions to transfer the amine group from glutamate onto oxaloacetate to form aspartate. Other Orobanchaceae species exhibit functional, though reduced, AAT activity. For example, *Striga* species and *Orobanche minor* exhibit about three-fourths and one-fourth, respectively, of the AAT activity in *Zea mays* [9]. *P. aegyptiaca* has several AAT homologs matching with known *A. thaliana* paralogs. The expression of the *P. aegyptiaca* homolog of AT4G31990 in *A. thaliana* correlates well with the changes in aspartate levels. We found multiple transcript isoforms for AAT and AS in transcriptome data of the two parasite species (Appendix A). Expression of these transcripts indicates some specialization for developmental stages in *P. aegyptiaca*, with one form associated with seedlings (stages 1 and 2), while another form was expressed predominantly during tubercle and later stages (Figure 8). The finding that expression patterns differed substantially between the two parasite species is consistent with the different styles of parasitism and levels of host dependence.

### 3.4. Parasite Effect on Host Plants

It is also interesting to ask whether the host root metabolite profiles changed upon parasitism. Our data indicate no change in *A. thaliana* roots parasitized by *P. aegyptiaca*, but *M. truncatula* roots parasitized by *T. versicolor* showed a 75% drop in asparagine levels (Figure 4). Asparagine is the most abundant amino acid in the *Medicago* phloem sap, comprising approximately 70% of the total amino acids [37,38]. The decrease in root amino acid levels may represent a means of defense. For the holoparasite *Orobanche foetida* grown on *Vicia faba*, a parasite tolerant line was found to decrease its phloem amino acid concentration to 53% that of the susceptible line in response to parasitism, and this was postulated to be a potential defense mechanism [3]. However, in our case, the *M. truncatula* host is still susceptible to parasitism, despite a dramatic decrease in asparagine levels. Decreased levels of amino acids in some hosts could also reflect the inability of these hosts to compensate for an additional substantial sink that parasitic plants represent. Further study is warranted to understand this response in infected hosts. 

## 4. Materials and Methods 

### 4.1. Plant Material and Sample Generation

Seeds of *P. aegyptiaca* were collected at Newe Na’ar Research Center, A.R.O. Israel (Courtesy of Dr. D. Joel). The seeds were from lot No. 32142, collected from *P. aegyptiaca* growing in a tomato field near Mavo Hama, Israel, in July 2009. Seeds of the outcrossing species *T. versicolor* were collected from an open pollinated population growing in a pasture land south of Napa California in 2015 (GPS location: 38°13’33.2”N, 122°16’11.7”W). A voucher specimen of *T. versicolor* from this collection was deposited in the Pennsylvania State University Herbarium. For the current experiments, both species were grown under controlled laboratory conditions as described in Yang et al. [21]. The stages of parasites collected centered on haustorial development and function, and corresponded to those defined as stages 1 through 4.2 by the Parasitic Plant Genome Project (Figure 1) [20,21]. The associated host roots were collected for all *T. versicolor* stages and for *P. aegyptiaca* stages 1, 3, and 4.1. 

### 4.2. Phelipanche aegyptiaca Growth on Arabidopsis thaliana

Parasite and host plants were grown in a polyethylene bag culture system [39]. The roots of 2-week-old *A. thaliana* plants were washed of soil and placed onto glass microfiber sheets (GF/A, Fisher Scientific) held in polyethylene bags. The bags were suspended vertically with the roots kept in the dark and shoots were exposed to short day conditions (10 h light, 14 h dark, approximately 100 µmol m^−2^ s^−1^ light ). Hoagland’s solution (¼ ×) was added to the bags as needed to maintain moisture. For stages not involving a host, identical conditions were used except that *A. thaliana* plants were absent from the bags. *P. aegyptiaca* seeds were surface sterilized with 1% (v/v) sodium hypochlorite for 8 min and rinsed thoroughly in sterile distilled water. Seeds were conditioned at room temperature (23 °C) for 6–7 days in the dark on moist glass microfiber disks in a Petri dish. The *P. aegyptiaca* seeds were stimulated to germinate by addition of GR-24 solution to create a final concentration of 1 ppm. The following day, *P. aegyptiaca* seeds were transferred to the *A. thaliana* roots or empty glass microfiber sheets in polyethylene bags using a fine brush. 

Samples consisting of either whole parasites or host roots were collected for metabolite analysis using a stereo-zoom microscope to enable precise dissection. For stage 1, seeds germinated in the absence of host roots were collected four days after GR-24 stimulation. Roots of non-infected *A. thaliana* were also collected at this stage. Stage 2 was also collected four days after GR-24 treatment as for stage 1, with the exception that the seedlings were placed on a mat of *A. thaliana* roots to expose them to haustorial-inducing factors for six hours prior to harvesting. Stage 3 parasites were placed on *A. thaliana* roots and were harvested seven days after GR-24 stimulation, at which time seedlings were attached to the host root, but did not show signs of swelling that would indicate vascular connections. For this stage, approximately 1 cm sections of host root were also collected, by dissecting them away from the associated parasites. Stage 4.1 parasite and host roots were collected at 10 days post GR-24 stimulation when the parasite had developed into a small tubercle. Stage 4.2 *P. aegyptiaca* was collected 14 days after GR-24 stimulation and was characterized by the development of adventitious roots on the tubercles. Host roots were not collected at this stage.

### 4.3. Triphysaria versicolor Growth on Medicago truncatula

*T. versicolor* seeds were surface sterilized with 1% (v/v) sodium hypochlorite, washed, suspended in water, and placed at 4 °C for two days. On the third day, the seeds were placed onto agar media containing ¼ × Hoagland’s solution in 0.6% Phytoagar and incubated for germination at 16 °C in a controlled environment chamber with 80% relative humidity and a 12 h light cycle. On day 14, *T. versicolor* seedlings were transferred onto plates containing ¼ × Hoagland’s media supplemented with 1% sucrose and placed in a 25 °C growth room under a 16 h light cycle. Two rows of 10–12 *T. versicolor* seedlings were arrayed along the top and middle of the plates. For metabolomics, *T. versicolor* seedlings were treated either with or without a *M. truncatula* host and harvested at indicated time points. For stage 1, *T. versicolor* and *M. truncatula* roots were harvested from their separate plates 22 days after germination of *T. versicolor*. For stages 2–4, *M. truncatula* seedlings were removed from their plates on day 22 and placed onto *T. versicolor* seedlings so that roots of *M. truncatula* seedlings were overlaid onto the ends of the *T. versicolor* roots. Stages 2, 3, and 4 were harvested 6, 24, and 72 h after the transfer of the *M. truncatula* seedlings, respectively. 

### 4.4. Tissue Harvesting

The small size of young parasites and need for painstaking dissection in harvesting presented challenges for tissue harvesting. Two experiments were conducted for *P. aegyptiaca*, with the first one involving harvesting by placing tissue immediately into UPLC grade methanol in microcentrifuge tubes held in a benchtop cooler rack pre-chilled to −20 °C. Samples were then transferred to −80 °C for storage. *T. versicolor* was also harvested this way. A second metabolomics analysis of *P. aegyptiaca* was conducted such that samples were collected directly into liquid nitrogen and held there until transfer to −80 °C for storage. For both methods, all samples were then freeze-dried and stored in a desiccator until metabolite profiling. 

### 4.5. Metabolite Extraction and Polar Metabolite Level Analyses

Metabolite extractions and analyses by gas chromatography-mass spectrometry (GC-MS) and Waters ultra-performance liquid chromatography (UPLC) were performed as previously described [40,41]. This untargeted GC-MS method allowed analysis of relative levels of 11 polar metabolites, including sugars, sugar alcohols and acids, carboxylic acids, and other compounds, while the targeted UPLC method allowed analysis of 17 amino acids and organic amines. We detected many additional metabolites and peaks, but because of differences in sample runs, we excluded all metabolites that were below the detection limit in at least one experiment. However, we were also interested in plant-specific presence–absence of metabolites, so metabolites that were not below the limit of detection for a particular sample type were included. For instance, host plants are known to be incapable to synthesize mannitol, but the parasites are able to make mannitol. Proteinogenic amino acids should be detectable. Some metabolites detected in *T. versicolor* were not detectable in *P. aegyptiaca* and *vice versa*, and they could contribute to species differences. It is likely that species differences lead to many of these differences in range of abundance in the more “obscure” polar metabolites. Because it was impossible to distinguish between the species specificity and the detection limit problems associated with batch effect, these metabolites were not included in statistical analyses. Briefly, polar metabolites were extracted from dry powdered samples using a modified biphasic Bligh and Dyer protocol [42], with 10 mM HCl as the aqueous solvent. For *P. aegyptiaca*, the starting material dry weights ranged from 0.79 to 1.99 mg and metabolites were extracted in 100 µL each chloroform and aqueous phase solvents. *T. versicolor* was extracted in varying quantities of solvent based on starting dry weight, normalizing to 1 mg dry weight per 100 µL aqueous phase solvent, in order to accommodate the large *M. truncatula* host root diameters relative to that of *T. versicolor*. For GC-MS analysis, 50 µL of the aqueous phase were dried under a stream of N_2_ gas, trimethylsilyl (TMS) derivatives of polar metabolites pretreated with methoxyamine HCl (MOX) were prepared, and 1 µL of a 30 µL derivatization mixture was injected on an Agilent 7890A-5975C series GC-MS with a 30-m DB-5MS-DG column (0.25 mm × 0.25 μm, Agilent Technologies, Santa Clara, CA, USA). For UPLC analysis, 5 µL of aqueous phase were used in a 25-µL AccQ-Tag^TM^ derivatization reaction and 0.5 µL were injected on an Acquity H-class UPLC system equipped with a fluorescence detector and analyzed according to the manufacturer’s recommendations using Waters 10.2 min method for analysis of cell culture amino acids (Waters, Milford, MA, USA). 

### 4.6. Metabolite-Level Data Processing

Data were collected from three separate experiments, two with *P. aegyptiaca* and one with *T. versicolor*. The data included absolute and relative levels of FAAs and polar metabolites (sugars, sugar alcohols, and carboxylic acids), respectively. Four to five biological replicates per developmental stage were done for each experiment. Each biological replicate represented a pooling of enough individual seedlings (3 to 140, depending on the stage) to reach approximately 1 mg of dry sample mass. For GC-MS data, compound identification and level quantification were done as previously described [40,41]. Briefly, three different complementary spectral libraries (NIST spectral library (National Institute of Standards and Technology, Gaithersburg, MD, USA)), FiehnLib spectral and retention time library [43], and an in-house generated spectral and retention time library using about 250 metabolite standards) were used to identify metabolites based on their spectra and retention times. Automated mass spectrometry deconvolution and identification system (AMDIS, NIST) was used to deconvolute signals from the coeluting compounds and to select specific fragments for each compound for quantitation. The enhanced mass selective detector ChemStation software version E.02.00.493 (Agilent Technologies, Santa Clara, CA, USA) was used in combination with the three above-mentioned libraries to obtain relative levels of polar metabolites. Manual curation was used on an individual basis (QEdit function of the ChemStation software version E.02.00.493) after automated peak area integration. The sugar alcohol ribitol was used as an internal standard to account for recovery. Peak areas of the multiply labeled internal standards were used to confirm the absence of matrix effects specific to particular categories of polar metabolites relative to ribitol ([U-^13^C_6_]-glucose was used for sugars and sugar phosphates and [2,2,4,4-D_4_]-citrate for carboxylic and organic acids). For UPLC data, standard curves were generated for each amino acid and used for amino acid identification and to quantify absolute levels of 17 amino acids. Norvaline was used as an internal standard to account for recovery. All samples were also standardized relative to the sample dry weight used for extractions. PCA was conducted on metabolite level correlations for all experiments together and for each host-parasite species set separately (JMP Pro 12 software, SAS, Cary, NC, USA). A 3D score plot was generated, which includes the data from all four species together. Raw uncorrected metabolomics data are provided in Appendix A. ANOVA was performed to evaluate the effect of stage, experimental batch, and species on each of the 28 metabolites. Three separate ANOVA tables were generated: (i) ANOVA was performed using the data from all three experiments and (ii) ANOVA was performed using data from each parasite/host system separately (Appendix A). Histograms of the *p*-values were plotted (Appendix A). ANOVA tables and histograms were generated using the MetabolomicsBasics package within R, as described by [44]. 

### 4.7. Split Root System

Sterilized *M. truncatula* seeds were germinated on ¼ × Hoagland’s nutrient plates containing 5 atom % ^15^N [95.66 µM 98 atom % ^15^N Ca(NO_3_)_2_], 1.15 mM Ca(NO_3_)_2_, 1.25 mM KNO_3_, 0.25 mM KH_2_PO_4_, 0.50 mM MgSO_4_, micronutrients, and 0.7% (w/v) Phyto agar. Such low ^15^N enrichment was needed for accurate subsequent stable isotope analysis by isotope-ratio mass spectrometer (IRMS) described below. One-week-old *M. truncatula* seedlings were transferred to rhizotrons and watered with 5 atom % ^15^N labeled ¼ × Hoagland’s media. Approximately three weeks later, *M. truncatula* plants with lateral roots in sufficient length were washed in deionized H_2_O and transplanted to split-root rhizotrons (Figure 6A). Roots in rhizotron A were watered with 5 atom % ^15^N labeled ¼ × Hoagland’s nutrient media and roots in rhizotron B were watered with H2O. One-week-old *T. versicolor* seedlings germinated on ¼ × Hoagland’s nutrient plates were aligned with *M. truncatula* in rhizotron B (Figure 6B). Samples of *M. truncatula* and *T. versicolor* were collected 12 days after alignment (Mt-Infected and Tv-Attached, respectively). Roots of *M. truncatula* growing in rhizotron B without *T. versicolor* were used as a control (Mt-Uninfected, Figure 6C). *T. versicolor* growing by itself in rhizotron watered with either H2O (Tv-Water) or ¼× Hoagland’s media (Tv-Hoagland’s) were collected as controls (Figure 6D). 

Plant material was freeze-dried and ground to a fine powder in a mortar cooled with liquid nitrogen. For each sample, 5–20 mg was submitted for ^15^N isotope analysis and amino acids of total proteins analysis. Stable isotope abundance analysis on ^15^N labeled amino acids was performed in the Stable Isotope Facility at UC Davis. Proteins were acid hydrolyzed with 6 M hydrochloric acid for 70 min at 105 °C, derivatized as N-acetyl isoprotyl esters, and analyzed on a Thermo Trace 1310 GC coupled to a Thermo Scientific Delta V Advantage IRMS(GC-C-IRMS). Arginine, histidine, and serine were not derivatized using this method. Tyrosine and threonine were below the limit of detection and the limit of quantification, respectively. Composition of amino acids of total proteins were analyzed at the Molecular Structural Facility, UC Davis. Proteins were acid hydrolyzed with 6 M hydrochloric acid for 24 h at 110 °C by L-8800 Hitachi amino acid analyzer, which separated amino acids by an ion-exchange column in HPLC. Total protein levels were calculated by adding up the levels of individual amino acids. Data for cysteine, methionine, and tryptophan were not available as these amino acids are destroyed during protein hydrolysis. These amino acids are present at trace levels in proteins and, therefore, not including them in total protein calculations should not significantly influence protein level estimations.

## 5. Conclusions

In summary, comparative metabolite profiling revealed that the metabolic response of parasitic Orobanchaceae to the availability of a host varies substantially between parasites of different nutritional dependency on their hosts. Our study showed that parasite metabolite profiles better reflected the developmental stage of the parasite than the resource availability of the host. This work also highlights the importance to the parasite of nitrogen acquisition from the host and supports this phenomenon as a potential target for control strategies against parasitism. More work is needed to understand precisely how parasites take nitrogen and amino acids from their hosts, and the extent to which amino acid levels and ratios have an effect on parasitism.

## Figures and Tables

**Figure 1 metabolites-09-00114-f001:**
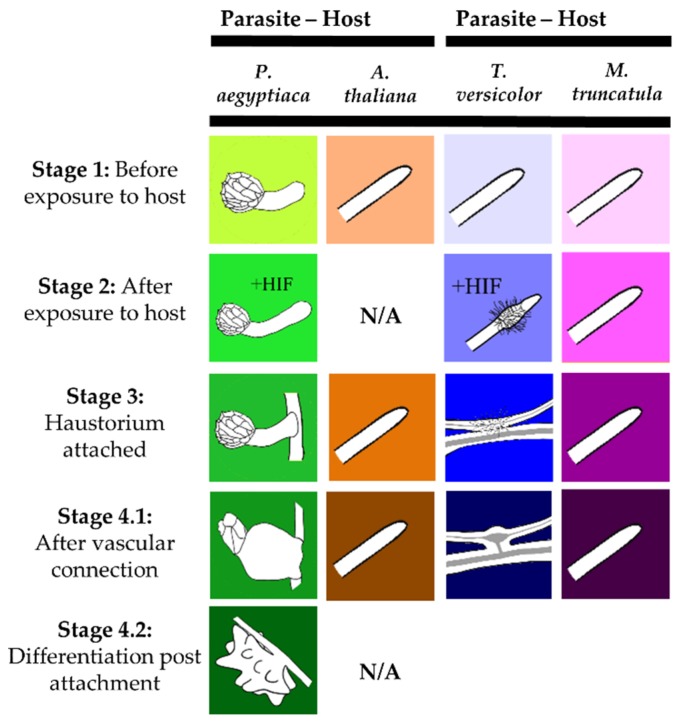
Stages of the Orobanchaceae species analyzed for metabolite profiles in this study. These stages correspond to those studied in previous transcriptome studies of Orobanchaceae [20,21]. Corresponding host roots of *A. thaliana* and *M. truncatula* were collected for each stage indicated for the parasites *P. aegyptiaca* and *T. versicolor*, respectively. Figure was adapted from Yang, Wafula, Honaas, Zhang, Das, Fernandez-Aparicio, Huang, Bandaranayake, Wu, Der, Clarke, Ralph, Landherr, Altman, Timko, Yoder, Westwood, and dePamphilis [21].

**Figure 2 metabolites-09-00114-f002:**
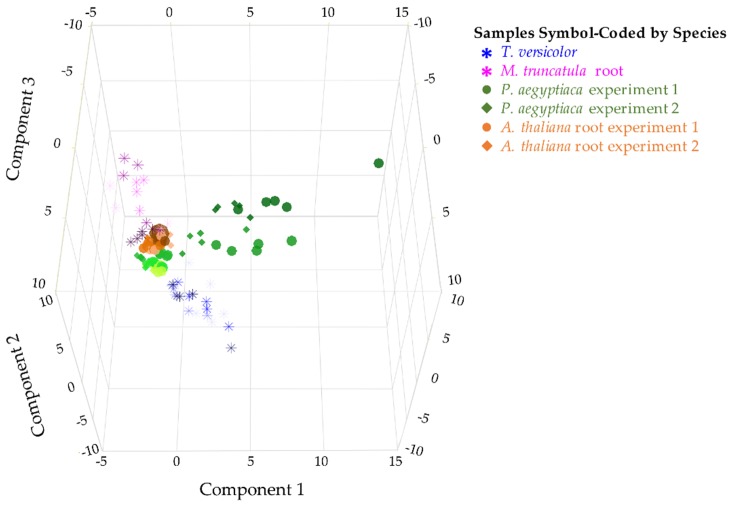
Comparison of two parasitic plant species and their associated hosts. PCA on metabolite level correlations. A 3D score plot of PC1, PC2, and PC3 is shown, accounting for 26.7%, 23.5%, and 15.1% of the variation between samples, respectively. PCA was performed on combined data from two experiments with five stages of *P. aegyptiaca* and three stages of *A. thaliana* host root samples (shown in green and orange, respectively, using circles to represent the first experiment and diamonds to represent the second) and one experiment with four stages of *T. versicolor* and four stages of *M. truncatula* host root samples (shown in blue asterisks and pink asterisks, respectively). Earlier time points are shown in lighter color shades. The data used relative levels of sugars, sugar alcohols, and carboxylic acids and absolute levels of free amino acids (FAAs) and included four to five biological replicates per developmental stage.

**Figure 3 metabolites-09-00114-f003:**
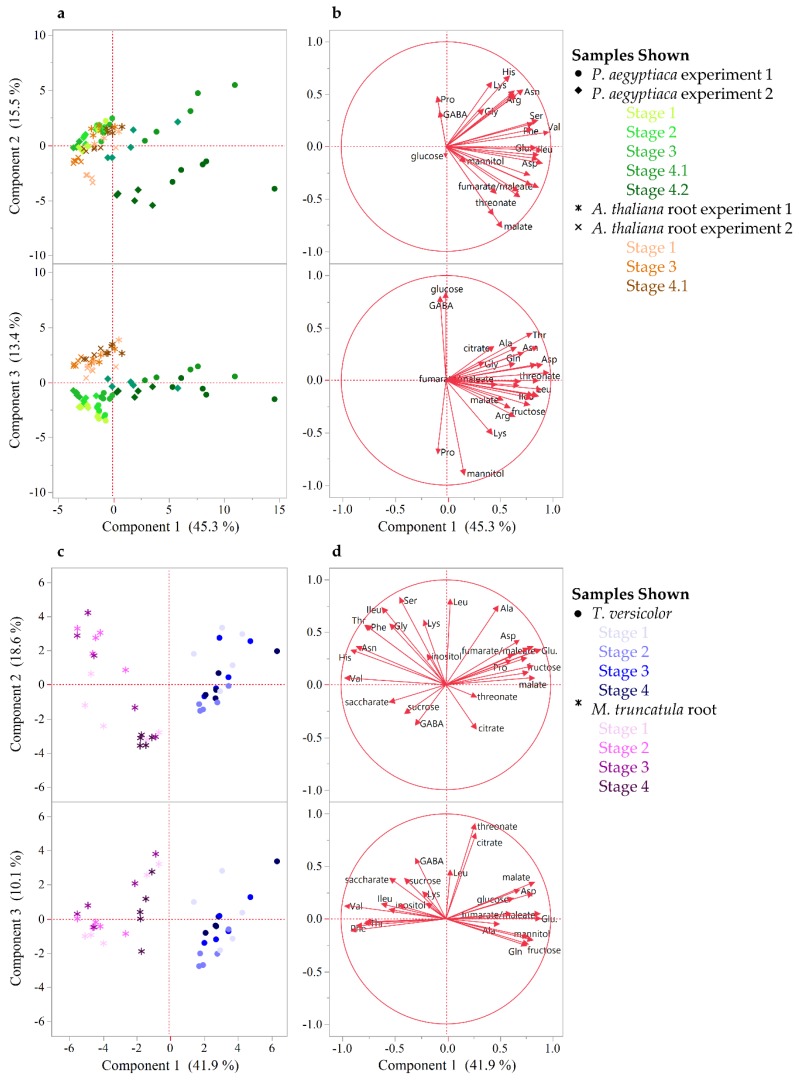
Metabolite profile comparison between developmental stages for host and parasite. (**a**,**b**) PCA on *P. aegyptiaca* and host metabolite level correlations. PCA was performed on combined data from two experiments with five stages of *P. aegyptiaca* and three stages of *A. thaliana* host root samples. The first three PCs are shown and together account for 74% of the variation between samples. (**c**,**d**) PCA on *T. versicolor* and host *M. truncatula* metabolite level correlations. The first three PCs are shown and together account for 71% of the variation between samples. (**a,c**) Score plots illustrate the separation between samples. (**b,d**) The loading plots illustrate the variables that account for the separation between samples. The data used relative or absolute levels of sugars and FAAs, respectively, and included four to five biological replicates per developmental stage.

**Figure 4 metabolites-09-00114-f004:**
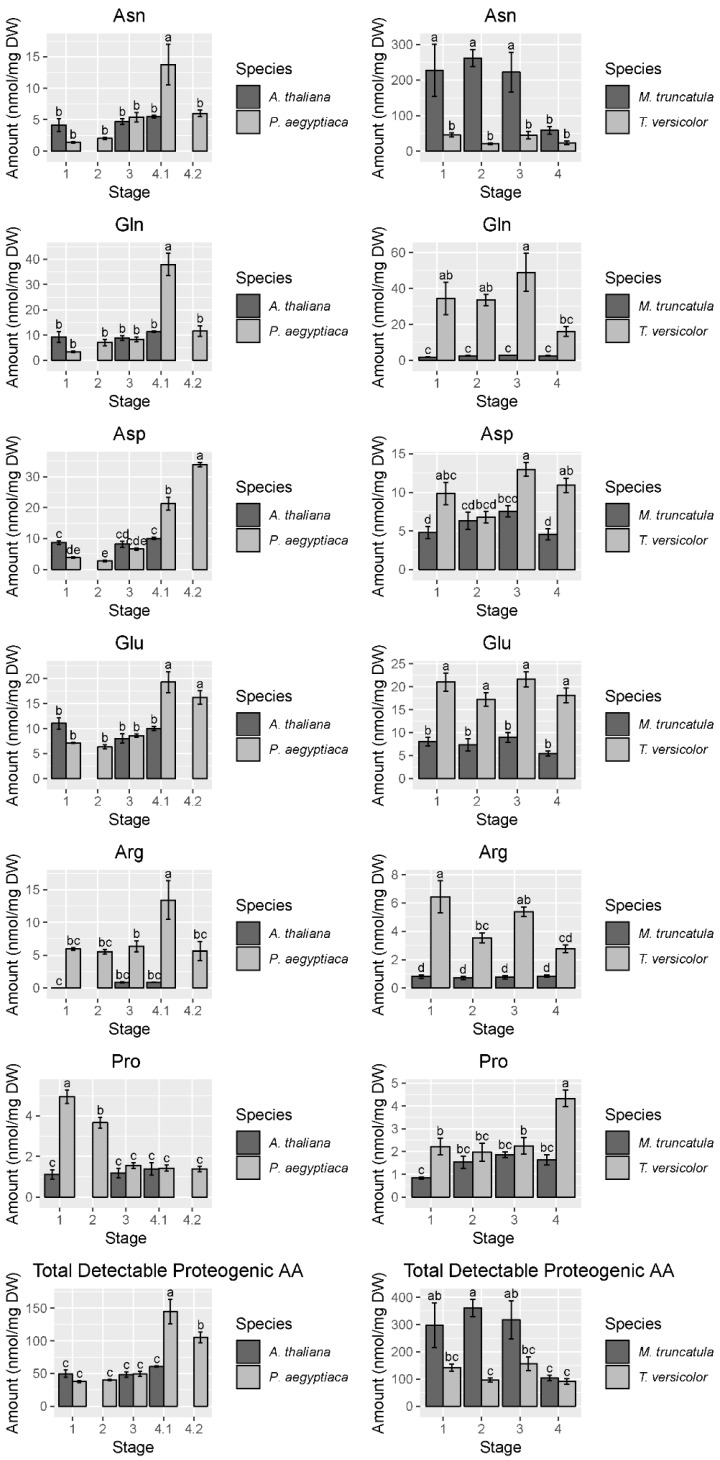
Abundance of selected free amino acids and total detectable protein-coding free amino acids. Data are taken from the first *P. aegyptiaca* experiment. Four to five biological replicates are used for each sample type. Note the difference in scale between the *P. aegyptiaca–A. thaliana* and the *T. versicolor–M. truncatula* systems, particularly the high levels of asparagine in *M. truncatula*. In cases where *A. thaliana* host root was not collected for corresponding *P. aegyptiaca* stages, no data are shown. Tukey HSD connecting letters are given, run on host and parasite sample values concurrently. Standard error bars are shown.

**Figure 5 metabolites-09-00114-f005:**
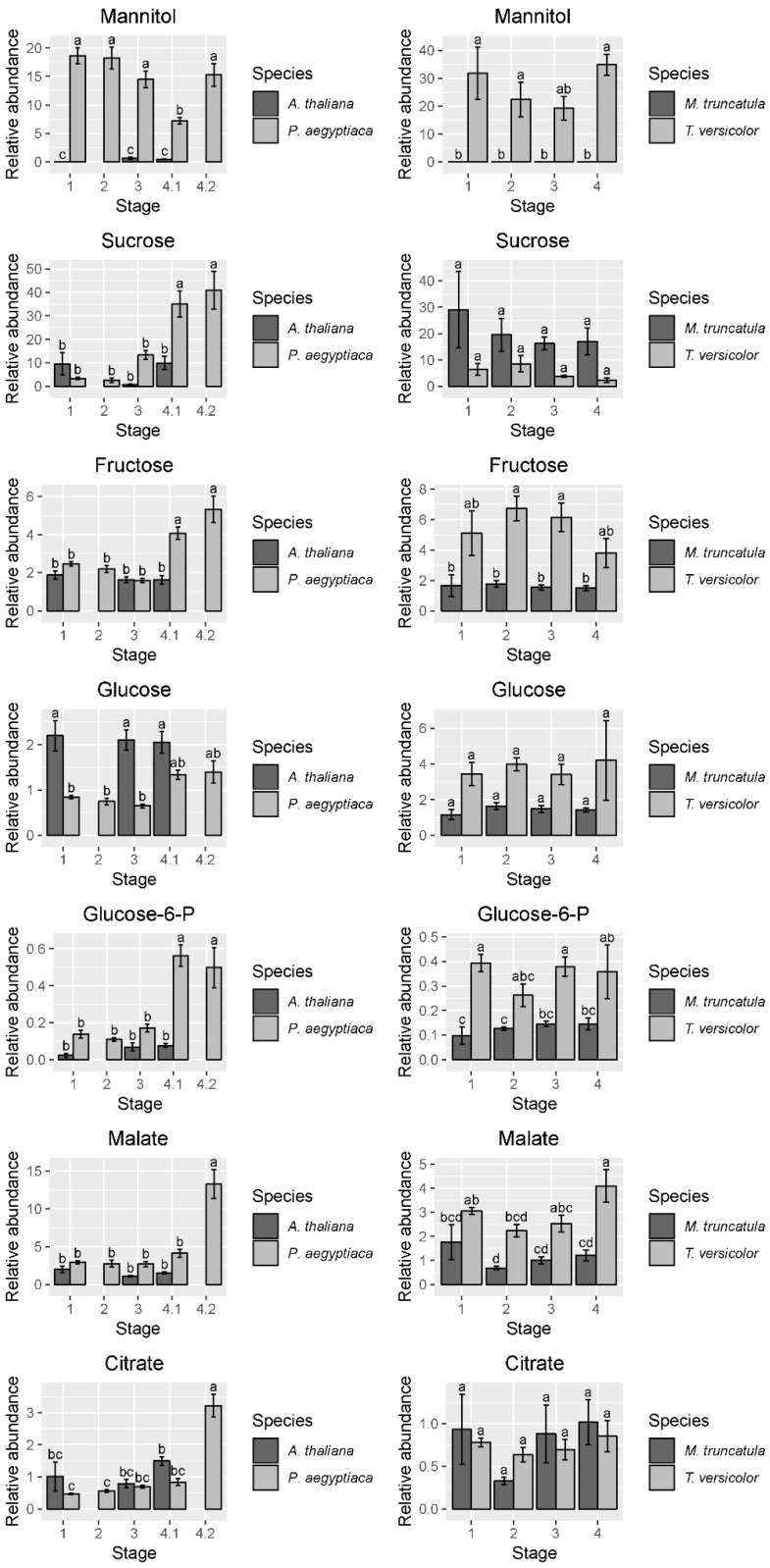
Abundance of selected sugars, sugar alcohols, and carboxylic acids. Data for graphs are from the first experiment in *P. aegyptiaca*. Four to five biological replicates are used for each sample type. Amounts were normalized from peak areas of the quantitation ion selected for each compound. In cases where *A. thaliana* host roots were not collected for the corresponding *P. aegyptiaca* stages, no data are shown. Tukey HSD connecting letters are given, run on host and parasite sample values concurrently. Standard error bars shown.

**Figure 6 metabolites-09-00114-f006:**
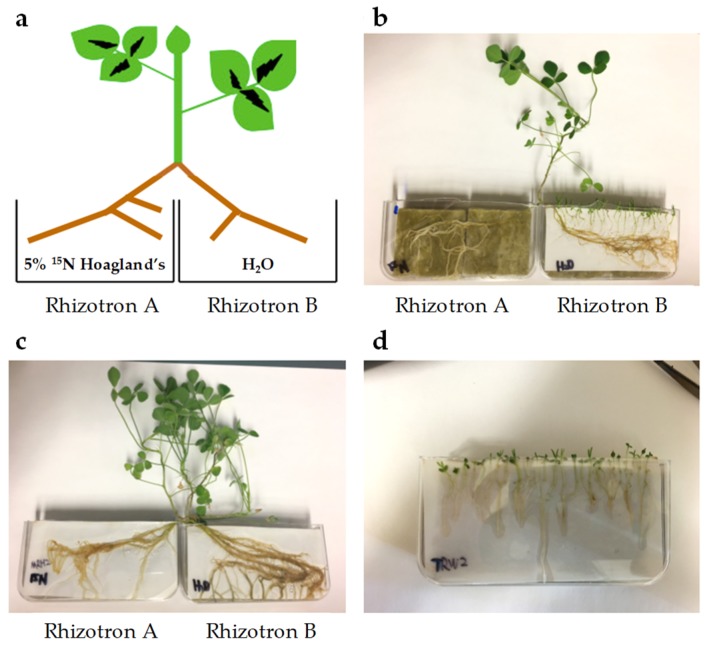
Split root system. (**a**) Diagram of split root system of 5 atom % ^15^N labelled *M. truncatula*. (**b**) *Triphysaria versicolor* aligned with *M. truncatula* in rhizotron B (Tv-Attached and Mt-Infected, respectively). (**c**) *M. truncatula* grown on split rhizotrons without *T. versicolor* (Mt-Uninfected). (**d**) *T. versicolor* grown in their own in rhizotron given either water only or ¼ × Hoagland’s solution (Tv-Water or Tv-Hoagland’s, respectively).

**Figure 7 metabolites-09-00114-f007:**
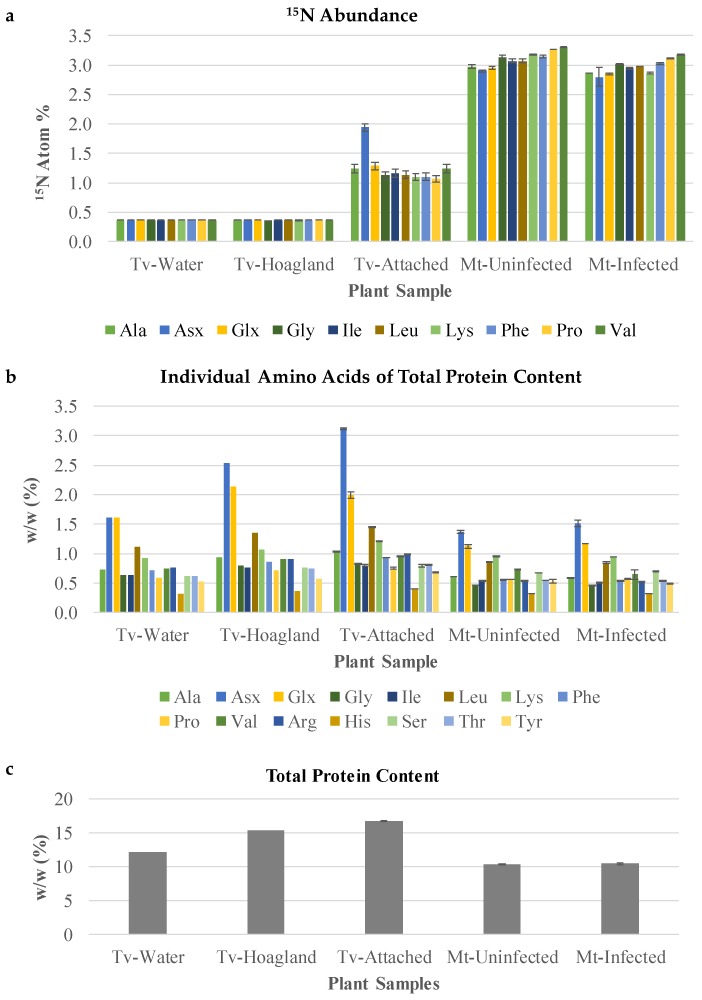
Amino acid analysis after protein hydrolysis treatment of split-root labeled *T. versicolor*. (**a**) An ^15^N abundance analysis of amino acids of samples collected from the split root system. Tv-Water and Tv-Hoagland’s represent parasites grown in the absence of host roots and ^15^N nitrate. As such, their amino acids are labeled to the level of 15N natural abundance, which is about 0.4. (**b**) Individual amino acids of total proteins content of the same samples used in ^15^N abundance analysis. (**c**) Total protein content, representing the sum of the individual amino acid contents. Mt-Uninfected, roots of *M. truncatula* growing in rhizotron B without *T. versicolor*. Mt-Infected, roots of *M. truncatula* growing in rhizotron B that had *T. versicolor* attached for 12 days of co-culture. Tv-Water, *T. versicolor* growing in a rhizotron on their own with H_2_O. Tv-Hoagland’s, *T. versicolor* growing in a rhizotron on their own with ¼ × Hoagland’s media. Tv-Attached, *T. versicolor* growing in rhizotron B collected after attachment on *M. truncatula*. Standard deviation bars shown.

**Figure 8 metabolites-09-00114-f008:**
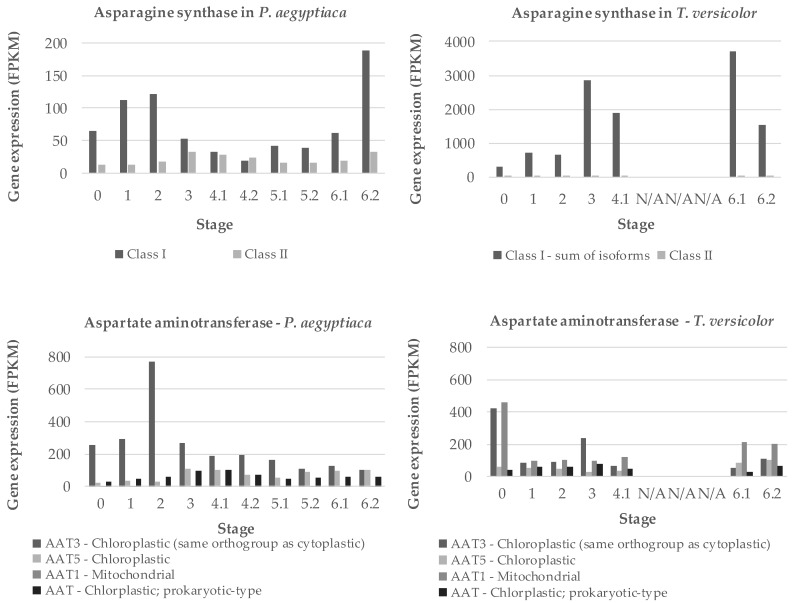
Expression of two amino acid metabolism genes. Gene expression of asparagine synthases (AS) class I and II and aspartate amino transferases (AAT) enzymes is shown for the obligate holoparasite *P. aegyptiaca* and facultative hemiparasite *T. versicolor*. Gene expression is shown in fragments per kilobase million (FPKM). Stages 5.1 and 5.2 represent shoots and roots, respectively, harvested before the parasite emerges from the soil. Stages 6.1 and 6.2 represent above-ground vegetative tissue and floral buds, respectively, after the parasite emerges from the soil. FPKM levels for *P. aegyptiaca* AAT5 and AAT1 and *T. versicolor* AAT3 and AAT1 are the sums of the FPKM levels for two or more bioinformatically identified isoforms of the same gene. The gene expression data are from previous study through the Parasitic Plant Genome Project [21], and contigs used are presented in Appendix A.

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
