# Peer review of "Comparative Metabolomics of Early Development of the Parasitic Plants Phelipanche aegyptiaca and Triphysaria versicolor"

_metabolites, 2019, doi:10.3390/metabo9060114_

Round 1
Reviewer 1 Report
Clermont et al. describe a series of experiments conducted to shed some light on the metabolism of two parasitic plants (holo and hemi) and their respective hosts. They quantified (in part absolutely) a set of 28 metabolites using GC and LC-MS based methods for different developmental stages during plant infection. Additionally, they conducted 15N-flux experiments and measured some gene expression for amino acid metabolism related genes. Due to the observable differences in parasite metabolic profiles following attachment they conclude that parasites maintain autonomy of their metabolism.
Comments and suggestions:
Figure2: I would recommend to substitute Figure 2. A 3D PCA scores plot is always difficult to grasp for the reader in 2D print representation, especially when many factors are involved which have to be coded with individual symbols and colours (stage, experiment, species). You get back to PCA effects in Figure 3 anyway. If your main point is to give an impression of the major sources of variance in your metabolic profiles why not calculating an ANOVA with the relevant factors (species, stage, experiment) for every metabolite and then plot the histograms for the 28 P-values (similar as in DOI: 10.1007/978-1-4939-7819-9_20).
Figure 3: Please check the symbols in the legend of a/b, I suppose the asterisks are A.thaliana
L132f: In the homoparasite part of your study you observe you observe nearly all your 28 metabolites to be systematically increased after stage 4, which explains the larger variance for these replicates but is counter intuitive (profiles highly similar prior to but different after attachment). While it may be correct it could hint at a systematic experimental bias too. Please check with your sample meta data if it is potentially associated with differences in sampling procedure, i.e. include the amount of extracted sample DW as a factor in your ANOVA and check P values. (I know that you corrected for differences by adjusting extraction volume but still…). Another possibility to check yourself is to get back to the GC-MS data. You only evaluated 11 metabolites out of this analysis but will have measured many more. If you find that also those signals which you did not include in the manuscript are relatively higher in stage 4.1/4.2 I would start to worry a little and consider a normalization step.
Figure 7: I did not get why 5 additional AAs are in (b) which are missing in (a), please explain briefly. You focus in your discussion on Tv-Attached, as this is the most interesting group. However, as you have excellent reproducibility here you may want to test if Mt-Uninfected is really not different from Mt-infected. On a per AA basis this probably is true but judging based on (a) I would guess that Mt-infected is overall systematically lower enriched.
280f: Can you be sure that the FAAs are host derived (only because you observe increased levels)? This is what you need to prove using 15N experiments as done for Mt, right? Or am I missing something?
General: Please make your metabolite data available (either in a public repository) or at least in a Supplemental Table to allow others a potential reuse.
Author Response
Response to reviewers.
Reviewer comments in black; Author’s relies in blue font.
Numbering refers to the revised version with the “All Markup” feature enabled.
Ethical statement added (as requested by the editors): lines 397 - 402
Seeds of P. aegyptiaca were collected at Newe Na’ar Research Center, A.R.O. Israel (Courtesy of Dr. D. Joel). The seeds were from lot No. 32142, collected from P. aegyptiaca growing in a tomato field near Mavo Hama, Israel, in July 2009. Seeds of the outcrossing species T. versicolor were collected from an open pollinated population growing in a pasture land south of Napa California in 2015 (GPS location: 38°13'33.2"N, 122°16'11.7”W). A voucher specimen of T. versicolor from this collection was deposited in the Pennsylvania State University Herbarium.
Open Review
(x) I would not like to sign my review report
( ) I would like to sign my review report
English language and style
( ) Extensive editing of English language and style required
( ) Moderate English changes required
(x) English language and style are fine/minor spell check required
( ) I don't feel qualified to judge about the English language and style
Yes | Can be improved | Must be improved | Not applicable | |
Does the introduction provide sufficient background and include all relevant references? | (x) | ( ) | ( ) | ( ) |
Is the research design appropriate? | (x) | ( ) | ( ) | ( ) |
Are the methods adequately described? | (x) | ( ) | ( ) | ( ) |
Are the results clearly presented? | (x) | ( ) | ( ) | ( ) |
Are the conclusions supported by the results? | (x) | ( ) | ( ) | ( ) |
Comments and Suggestions for Authors
Clermont et al. describe a series of experiments conducted to shed some light on the metabolism of two parasitic plants (holo and hemi) and their respective hosts. They quantified (in part absolutely) a set of 28 metabolites using GC and LC-MS based methods for different developmental stages during plant infection. Additionally, they conducted 15N-flux experiments and measured some gene expression for amino acid metabolism related genes. Due to the observable differences in parasite metabolic profiles following attachment they conclude that parasites maintain autonomy of their metabolism.
Comments and suggestions:
Figure2: I would recommend to substitute Figure 2. A 3D PCA scores plot is always difficult to grasp for the reader in 2D print representation, especially when many factors are involved which have to be coded with individual symbols and colours (stage, experiment, species). You get back to PCA effects in Figure 3 anyway. If your main point is to give an impression of the major sources of variance in your metabolic profiles why not calculating an ANOVA with the relevant factors (species, stage, experiment) for every metabolite and then plot the histograms for the 28 P-values (similar as in DOI: 10.1007/978-1-4939-7819-9_20).
Great suggestion. We have calculated an ANOVA using the factors species, stage, and experimental batch and plotted histograms for the p-values using the “MetabolomicsBasics” R package with the Jaeger 2018 book chapter as a model, as suggested.
We have included three separate ANOVA tables: one with all data from the three experimental batches (two from P. aegyptiaca and one from T. versicolor); one with data from the two P. aegyptiaca experimental batches only; and one with data from the T. versicolor experimental batch (experimental batch is excluded as a factor in the last of the three ANOVAs). The ANOVA tables and histograms of p-values are in Table S1 and Figure S1, respectively. These are referenced in the following text in lines 517 – 523:
ANOVA was performed to evaluate the effect of stage, experimental batch, and species on each of the 28 metabolites. Three separate ANOVA tables were generated: (i) ANOVA was performed using the data from all three experiments and (ii) ANOVA was performed using data from each parasite/host system separately (Table S1). Histograms of the p-values were plotted (Figure S1). ANOVA tables and histograms were generated using the MetabolomicsBasics package within R as described by [44].
In results, ANOVA results are described in several places along with the relevant PCA results:
Lines 108 – 111: When all stages of the two parasite and two host species were compared on a combined basis, each species demonstrated a distinctive metabolite profile as determined by Principal Component Analysis (PCA) (Figure 2) and analysis of variance (ANOVA) (Figure S1, Table S1).
Lines 122 – 125: For P. aegyptiaca, the PCA and ANOVA indicates that metabolite profiles are distinguishable by developmental stage (Figure S1, Table S1), with pronounced differences separating stages before and after vascular linkage to the host (Stages 1-3 vs. Stages 4.1-4.2) (Figure 3A).
Lines 186 – 188: ANOVA results corroborate the PCA results, as all p-values for all tested metabolites with growth stage as a factor were found statistically insignificant, as they exceeded the value of 0.05 (Figure S1, Table S1).
We propose to keep the 3D PCA figure because it creates a visual impression such that, at this particular angle, the 3D allows to visualize sample separation given by the three principal components at the same time. We supplemented this figure with an ANOVA table and histograms of p-values, as suggested by the reviewer.
Figure 3: Please check the symbols in the legend of a/b, I suppose the asterisks are A. thaliana.
Thank you for pointing this out. The symbols have been corrected.
L132f: In the homoparasite part of your study you observe you observe nearly all your 28 metabolites to be systematically increased after stage 4, which explains the larger variance for these replicates but is counter intuitive (profiles highly similar prior to but different after attachment). While it may be correct it could hint at a systematic experimental bias too. Please check with your sample meta data if it is potentially associated with differences in sampling procedure, i.e. include the amount of extracted sample DW as a factor in your ANOVA and check P values. (I know that you corrected for differences by adjusting extraction volume but still…). Another possibility to check yourself is to get back to the GC-MS data. You only evaluated 11 metabolites out of this analysis but will have measured many more. If you find that also those signals which you did not include in the manuscript are relatively higher in stage 4.1/4.2 I would start to worry a little and consider a normalization step.
We agree that this is a valid concern. However, because the levels of several metabolites (Gly, glucose, mannitol, Lys, and His) did not change or even decreased (Pro) in both P. aegyptiaca experiments following the host attachment (stage 4), one can argue against this type of bias. In addition, the trends were reproducible for these metabolites in both experiments. The relevant data is presented in Supplemental Figure 2. The text about metabolites that did not change was already in the manuscript (e.g., lines 133 – 134).
Figure 7: I did not get why 5 additional AAs are in (b) which are missing in (a), please explain briefly. You focus in your discussion on Tv-Attached, as this is the most interesting group. However, as you have excellent reproducibility here you may want to test if Mt-Uninfected is really not different from Mt-infected. On a per AA basis this probably is true but judging based on (a) I would guess that Mt-infected is overall systematically lower enriched.
The two amino acid analyses were performed using different methods. Thanks for pointing out that this was not well differentiated in the text.
(1) For the amino acids of total protein analysis, standard vacuum hydrolysis procedure employs 6 M hydrochloric acid for 24 hours at 110° C. Samples were analyzed by L-8800 Hitachi amino acid analyzer, which separated amino acids by an ion-exchange column in HPLC.
(2) For 15N abundance assay, samples were treated with 6 M hydrochloric acid for 70 mins at 150 °C and then derivatized to N-acetyl amino acid isopropyl esters for GC-C-IRMS analysis. Arg, His, and Ser were not derivatized using this method. Tyr was below the limit of quantification for all of our samples and Thr was below LOQ for many samples, so those were not included in the 15N abundance analysis as well.
The text at line 535 – 550 has been updated as follows:
Plant material was freeze-dried and ground to a fine powder in a mortar cooled with liquid nitrogen. For each sample, 5 – 20 mg were submitted for 15N isotope analysis and amino acids of total proteins analysis. Stable isotope abundance analysis on 15N labeled amino acids was performed in the Stable Isotope Facility at UC Davis. Proteins were acid hydrolyzed with 6 M hydrochloric acid for 70 min at 105° C , derivatized as N-acetyl isoprotyl esters, and analyzed on a Thermo Trace GC 1310 GC coupled to a Thermo Scientific Delta V Advantage IRMS (GC-C-IRMS). Arginine, histidine, and serine were not derivatized using this method. Tyrosine and threonine were below the limit of detection and the limit of quantification, respectively. Composition of amino acids of total proteins were analyzed at the Molecular Structural Facility, UC Davis. Proteins were acid hydrolyzed with 6 M hydrochloric acid for 24 hr at 110° C by L-8800 Hitachi amino acid analyzer, which separated amino acids by an ion-exchange column in HPLC. Total protein levels were calculated by adding up the levels of individual amino acids. Data for cysteine, methionine and tryptophan were not available as these amino acids are destroyed during acidic protein hydrolysis. These amino acids are present at trace levels in proteins and therefore, not including them in total protein calculations should not significantly influence protein level estimations.
280f: Can you be sure that the FAAs are host derived (only because you observe increased levels)? This is what you need to prove using 15N experiments as done for Mt, right? Or am I missing something?
This is an excellent point and we addressed it in lines 285-292. Basically, the P. aegyptiaca seeds do not have much seed storage compounds due to their very small size, and after they germinate, they must attach to the host in order to survive. As such, the bulk of carbon and nitrogen has to come from the host after the attachment. T. vesicolor, on the other hand, can grow without the host, so the 15N experiment was required. However, we still do not know in what form these carbon and nitrogen sources are transported from the host to the parasite, so when we measured the levels of metabolites in the parasite, it was for a mixture of metabolites that have been transported from the host as well as those already metabolized by the parasite. Unfortunately, even 15N experiments are not able to distinguish what was transported vs. already metabolized. These experiments would only show that 15N was derived from the host. A carefully designed 15N time-course experiment could point to (based on the timing of 15N amino acid appearance in the parasite) bulk forms of N that are transported, but it still would not yield conclusive results since amino acid metabolism is extremely fast. It is also important to note that even the steady-state 15N experiment with T. versicolor was extremely difficult to perform. To discern the forms of N sources that are transported will require identification and characterization of the relevant transporters, which is a study on its own. The edited text now reads as follows:
As an obligate holoparasite, P. aegyptiaca is not able to grow without a host and completely relies on influx of host-derived metabolites such as sugars and amino acids. Therefore, the peak in total detectable proteogenic FAAs at stage 4.1 (Figure 4) likely reflected a strong influx of an unknown form of nitrogen from the host following access to host-derived nutrients. Similarly, these tubercles (stages 4.1 and 4.2) also had substantial general increases in sugar, sugar alcohol, and carboxylic acid levels, except for glucose and mannitol (Figure 5 and Figure S1). However, the specific host-derived metabolites that are transported to these tubercules remain to be uncovered.
General: Please make your metabolite data available (either in a public repository) or at least in a Supplemental Table to allow others a potential reuse.
A Supplemental Table with the raw metabolomics data has been added to the manuscript submission, in Table S3. Table S3 is referenced in line 512 of the text:
Metabolomics data is now provided in Table S3.

Reviewer 2 Report
The present work makes an overview on some compounds from primary metabolism, and the content and metabolism of C and N. The system is plant-plant root interaction, between a parasitic plant and its host. For the experiments two different plants with two different parasites have been used. The general conclusion is mostly based in the relevance of two amino acids in one particular interaction.
The study can be interesting, nevertheless I have some comments to the authors:
- Mostly, the experimental design is a bit confusing. When reading the paper there is no specific or clear objective in the work. Why are used two different parasites and two different hosts? It is difficult to compare when each interaction may behave in different way.
- What is biological replicate for the authors? The fact that in one case there are two experiments and in the other interaction, only one, makes me think that indeed, the replicates are more technical. There is an advice in metabolomics which is to make at least three different experiments to have three biological replicates and make technical for such analysis, at least in non-targeted metabolomics. The experiments need to be repeated and treated in the same way.
- How has been performed the identification of compounds? Is it targeted? untargeted? The quantification has been done how? Some more explanation should be presented in order to make easy to readers the replication of your experimental system.
- There are some mistakes in the writing. Please check the figure legends because in some cases do not match with the figure (specifically the symbols); check fig 3 for example. There are also some symbols like stars, that I don't see the match in the figure legend.
Author Response
Response to reviewers.
Reviewer comments in black; Author’s relies in blue font.
Numbering refers to the revised version with the “All Markup” feature enabled.
Ethical statement added (as requested by the editors): lines 397 - 402
Seeds of P. aegyptiaca were collected at Newe Na’ar Research Center, A.R.O. Israel (Courtesy of Dr. D. Joel). The seeds were from lot No. 32142, collected from P. aegyptiaca growing in a tomato field near Mavo Hama, Israel, in July 2009. Seeds of the outcrossing species T. versicolor were collected from an open pollinated population growing in a pasture land south of Napa California in 2015 (GPS location: 38°13'33.2"N, 122°16'11.7”W). A voucher specimen of T. versicolor from this collection was deposited in the Pennsylvania State University Herbarium.
Open Review
(x) I would not like to sign my review report
( ) I would like to sign my review report
English language and style
( ) Extensive editing of English language and style required
( ) Moderate English changes required
( ) English language and style are fine/minor spell check required
(x) I don't feel qualified to judge about the English language and style
Yes | Can be improved | Must be improved | Not applicable | |
Does the introduction provide sufficient background and include all relevant references? | (x) | ( ) | ( ) | ( ) |
Is the research design appropriate? | (x) | ( ) | ( ) | ( ) |
Are the methods adequately described? | (x) | ( ) | ( ) | ( ) |
Are the results clearly presented? | ( ) | (x) | ( ) | ( ) |
Are the conclusions supported by the results? | (x) | ( ) | ( ) | ( ) |
Comments and Suggestions for Authors
The present work makes an overview on some compounds from primary metabolism, and the content and metabolism of C and N. The system is plant-plant root interaction, between a parasitic plant and its host. For the experiments two different plants with two different parasites have been used. The general conclusion is mostly based in the relevance of two amino acids in one particular interaction.
The study can be interesting, nevertheless I have some comments to the authors:
- Mostly, the experimental design is a bit confusing. When reading the paper there is no specific or clear objective in the work. Why are used two different parasites and two different hosts? It is difficult to compare when each interaction may behave in different way.
The comparison between different parasites (with different hosts) was one of the main objectives of the research, although it seems it was not sufficiently explained. To improve this, we have rewritten a paragraph in the introduction to better explain the goals and design of the study. This is on lines 82-97 of the revised manuscript:
The objective of this research was to fill these knowledge gaps by characterizing parasitic plant metabolism across key developmental stages in their life cycle and by comparing related species that contrast in level of host dependence. For the comparison of developmental stages, we focused on the transition from free-living to host-dependent stages of their life cycles. The free-living stage occurs in early development when the parasites must grow and develop either autotrophically (for facultative parasites) or as seedlings using only seed-storage reserves (for holoparasites). The process of host attachment is therefore pivotal in that this transition marks the switch from independent metabolism to host-fed metabolism and comprises the essence of parasitism. For evolutionary comparisons, we used the facultative hemiparasite Triphysaria versicolor and the obligate holoparasite Phelipanche aegyptiaca, both members of the family Orobanchaceae, the only parasitic plant family that includes species that span all levels of host dependence [16]. Our goal was to determine whether these species would show evidence of shared metabolic processes that could indicate a cross-cutting strategy for parasite metabolism. Although we used different hosts for each parasite species (justified in more detail in [17]), we were able to contrast the different parasite species with their hosts and with each other to gain insight into the extent to which parasite metabolism is self-regulating.
We agree that it is difficult to compare very different plants, but we are doing this in a general way. In fact, most of the literature on parasitic plants features studies that look at only one parasite-host interaction and then make broad generalizations about all parasitic plants. Here we are trying specifically to compare two systems using similar methodologies so we can begin to appreciate how different parasite species can be, even if they are in the same family.
- What is biological replicate for the authors? The fact that in one case there are two experiments and in the other interaction, only one, makes me think that indeed, the replicates are more technical. There is an advice in metabolomics which is to make at least three different experiments to have three biological replicates and make technical for such analysis, at least in non-targeted metabolomics. The experiments need to be repeated and treated in the same way.
We agree that, normally, at least 3 separate experiments should be performed for the metabolomics. However, obtaining enough parasite tissue for analysis was a technical challenge as the parasites are small (especially P. aegyptiaca) and the separation of parasite and host is very laborious. Because P. aegyptiaca harvesting was so difficult, we conducted several preliminary experiments and 2 full experiments with this species to work out and confirm the methodology. Once this was set, it was easier to use the same approach for T. versicolor. Given our experience with P. aegyptiaca, and our ability to achieve consistent results with this species, (confirmed in 2 independent experiments, Table S1), we were satisfied with data from T. versicolor, which seems to have more stable metabolite profiles.
Within each experiment (regardless of the parasite), multiple individuals were pooled for each sample (see table below for P. aegyptiaca for numbers), and each sample came from different rhizotron(s) where parasites were growing on host plant roots. The pooled individuals in each sample were considered a biological replicate, representing the variability in the tissues/stage, and each of these was repeated to obtain 4 – 5 biological replicates. There were no technical replicates because the variance is so much smaller in technical than in biological replicates. So, each sample type had 8 – 10 biological replicates for P. aegyptiaca (considering the two experiments) and 4 – 5 biological replicates for T. vesicolor.
Stage Individuals/rep
Stage 1-3 ~140
Stage 4.1 ~55
Stage 4.2 ~3
The relevant text can be found in lines 486 – 491:
Data were collected from three separate experiments, two with P. aegyptiaca and one with T. versicolor. The data included absolute and relative levels of FAAs and polar metabolites (sugars, sugar alcohols, and carboxylic acids), respectively. Four to five biological replicates per developmental stage were done for each experiment. Each biological replicate represented a pooling of enough individual seedlings (3 – 140, depending on the stage) to reach approximately 1 mg of dry sample mass.
- How has been performed the identification of compounds? Is it targeted? untargeted? The quantification has been done how? Some more explanation should be presented in order to make easy to readers the replication of your experimental system.
We clarified this issue and added appropriate text in lines 458 – 463 (targeted vs. untargeted) and 480 – 497 (compound identification and data analysis). The metabolomics was a mixture of targeted (amino acid analysis by UPLC) and untargeted (other major polar metabolites by GC-MS) analyses. This is the edited text:
Metabolite extractions and analyses by gas chromatography-mass spectrometry (GC-MS) and Waters ultra-performance liquid chromatography (UPLC) were performed as previously described [40,41]. This untargeted GC-MS method allowed analysis of relative levels of 11 polar metabolites, including sugars, sugar alcohols and acids, carboxylic acids, and other compounds, while the targeted UPLC method allowed analysis of 17 amino acids and organic amines. We detected many additional metabolites and peaks, but because of differences in sample runs, we excluded all metabolites that were below the detection limit in at least one experiment.
We already published detailed methods for these analyses including identification of the compounds in doi: 10.1021/ac9019522, so originally we did not think it was necessary to include details in this manuscript. As requested by the reviewer, this time we also provided the requested information on identification of compounds and data analysis. New text was added to the 4.5. Metabolite Level Data Processing section of the Methods (lines 491 – 509):
For GC-MS data, compound identification and level quantification were done as previously described [37,38]. Briefly, three different complementary spectral libraries (NIST spectral library (National Institute of Standards and Technology, Gaithersburg, MD), FiehnLib spectral and retention time library [40], and an in-house generated spectral and retention time library using about 250 metabolite standards) were used to identify metabolites based on their spectra and retention times. Automated Mass Spectrometry Deconvolution and Identification System (AMDIS, NIST) was used to deconvolute signals from the coeluting compounds and to select specific fragments for each compound for quantitation. The Enhanced Mass Selective Detector ChemStation software (Agilent Technologies) was used in combination with the three above-mentioned libraries to obtain relative levels of polar metabolites. Manual curation was used on an individual basis (QEdit function of the ChemStation software) after automated peak area integration. The sugar alcohol ribitol was used as an internal standard to account for recovery. Peak areas of the multiply labeled internal standards were used to confirm the absence of matrix effects specific to particular categories of polar metabolites relative to ribitol ([U‐13C6]‐glucose was used for sugars, sugar alcohols, and sugar phosphates and [2,2,4,4‐D4]‐citrate for carboxylic and organic acids). For UPLC data, standard curves were generated for each amino acid and used for amino acid identification and to quantify absolute levels of 17 amino acids. Norvaline was used as an internal standard to account for recovery. All samples were also standardized relative to the sample dry weight used for extractions.
- There are some mistakes in the writing. Please check the figure legends because in some cases do not match with the figure (specifically the symbols); check fig 3 for example. There are also some symbols like stars, that I don't see the match in the figure legend.
Thank you for pointing this out. The symbols have been corrected.

Round 2
Reviewer 2 Report
The manuscript has gained in reading level. Also the justification of what the authors understand for biological replicate and explaining the experimental system, the publication can be published in that way.